# CAN A FRUIT FLY LEARN WORD EMBEDDINGS?

**Yuchen Liang** *
RPI
MIT-IBM Watson AI Lab
liangy7@rpi.edu

**Chaitanya K. Ryali**
Department of CS
UC San Diego
rckrishn@eng.ucsd.edu

**Benjamin Hoover**
MIT-IBM Watson AI Lab
IBM Research
benjamin.hoover@ibm.com

**Leopold Grinberg**
IBM Research
lgrinbe@ibm.com

**Saket Navlakha**
Cold Spring Harbor Laboratory
navlakha@cshl.edu

**Mohammed J. Zaki**
Department of CS
RPI
zaki@cs.rpi.edu

**Dmitry Krotov**
MIT-IBM Watson AI Lab
IBM Research
krotov@ibm.com

## ABSTRACT

The mushroom body of the fruit fly brain is one of the best studied systems in neuroscience. At its core it consists of a population of Kenyon cells, which receive inputs from multiple sensory modalities. These cells are inhibited by the anterior paired lateral neuron, thus creating a sparse high dimensional representation of the inputs. In this work we study a mathematical formalization of this network motif and apply it to learning the correlational structure between words and their context in a corpus of unstructured text, a common natural language processing (NLP) task. We show that this network can learn semantic representations of words and can generate both static and context-dependent word embeddings. Unlike conventional methods (e.g., BERT, GloVe) that use dense representations for word embedding, our algorithm encodes semantic meaning of words and their context in the form of sparse binary hash codes. The quality of the learned representations is evaluated on word similarity analysis, word-sense disambiguation, and document classification. It is shown that not only can the fruit fly network motif achieve performance comparable to existing methods in NLP, but, additionally, it uses only a fraction of the computational resources (shorter training time and smaller memory footprint).

## 1 INTRODUCTION

Deep learning has made tremendous advances in computer vision, natural language processing and many other areas. While taking high-level inspiration from biology, the current generation of deep learning methods are not necessarily biologically realistic. This raises the question whether biological systems can further inform the development of new network architectures and learning algorithms that can lead to competitive performance on machine learning tasks or offer additional insights into intelligent behavior. Our work is inspired by this motivation. We study a well-established neurobiological network motif from the fruit fly brain and investigate the possibility of reusing it for solving common machine learning tasks in NLP. We consider this exercise as a toy model example illustrating the possibility of "reprogramming" of naturally occurring algorithms and behaviors (clustering combinations of input stimuli from olfaction, vision, and thermo-hydro sensory system) into a target algorithm of interest (learning word embeddings from raw text) that the original biological organism does not naturally engage in.

The mushroom body (MB) is a major area of the brain responsible for processing of sensory information in fruit flies. It receives inputs from a set of projection neurons (PN) conveying information

---

*Yuchen Liang is an AI Horizons Scholar, part of the Rensselaer-IBM AI Research Collaboration (AIRC).

from several sensory modalities. The major modality is olfaction [2], but there are also inputs from the PN responsible for sensing temperature and humidity [29], as well as visual inputs [45; 6]. These sensory inputs are forwarded to a population of approximately 2000 Kenyon cells (KCs) through a set of synaptic weights [26]. KCs are reciprocally connected through an anterior paired lateral (APL) neuron, which sends a strong inhibitory signal back to KCs. This recurrent network effectively implements winner-takes-all competition between KCs, and silences all but a small fraction of top activated neurons [8]. This is the network motif that we study in this paper; its schematic is shown in Fig. 1. KCs also send their outputs to mushroom body output neurons (MBONs), but this part of the MB network is not included into our mathematical model.

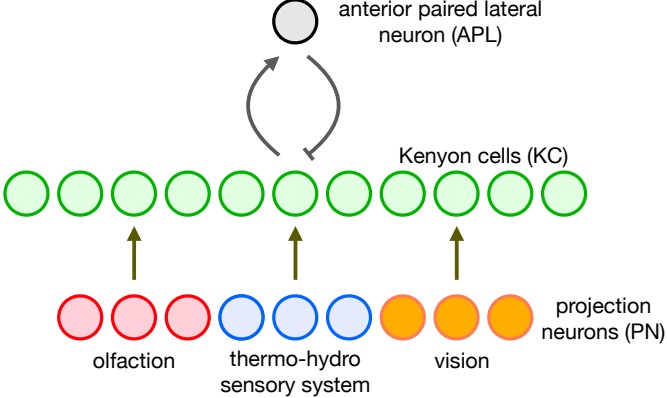

Figure 1: Network architecture. Several groups of PNs corresponding to different modalities send their activities to the layer of KCs, which are inhibited through the reciprocal connections to the APL neuron.

Behaviorally, it is important for a fruit fly to distinguish sensory stimuli, e.g., different odors. If a fruit fly senses a smell associated with danger, it's best to avoid it; if it smells food, the fruit fly might want to approach it. The network motif shown in Fig. 1 is believed to be responsible for clustering sensory stimuli so that similar stimuli elicit similar patterns of neural responses at the level of KCs to allow generalization, while distinct stimuli result in different neural responses, to allow discrimination. Importantly, this biological network has evolved to accomplish this task in a very efficient way.

In computational linguistics there is a long tradition [19] of using distributional properties of linguistic units for quantifying semantic similarities between them, as summarized in the famous quote by JR Firth: "a word is characterized by the company it keeps" [14]. This idea has led to powerful tools such as Latent Semantic Analysis [9], topic modelling [3], and language models like word2vec [30], GloVe [34], and, more recently, BERT [10] which relies on the Transformer model [44]. Specifically word2vec models are trained to maximize the likelihood of a word given its context, GloVe models utilize global word-word co-occurence statistics, and BERT uses a deep neural network with attention to predict masked words (and the next sentence). As such, all these methods utilize the correlations between individual words and their context in order to learn useful word embeddings.

In our work we ask the following question: can the correlations between words and their contexts be extracted from raw text by the biological network of KCs, shown in Fig. 1? Further, how do the word representations learned by KCs differ from those obtained by existing NLP methods? Although this network has evolved to process sensory stimuli from olfaction and other modalities and not to "understand" language it uses a general purpose algorithm to embed inputs (from different modalities) into a high dimensional space with several desirable properties, which we discuss below.

Our approach relies on a recent proposal that the recurrent network of mutually inhibited KCs can be used as a "biological" model for generating sparse binary hash codes for the input data presented at the projection neuron layer [8]. It was argued that a matrix of random weights projecting from PN layer into the KCs layer leads to the highly desirable property of making the generated hash codes locality sensitive, i.e., placing similar inputs close to each other in the embedding space and pushing distinct stimuli far apart. A subsequent study [39] has demonstrated that the locality sensitivity of the hash codes can be significantly increased, compared to the random case, if the matrix of weights from PN to KCs is learned from data. The idea of using the network of KCs with random projections for NLP tasks has also been previously explored in [37], see discussion in section 6.

Biologically, there is an ongoing debate in the neuroscience community regarding whether these projections are random. For instance, [5] argues for the random model, while [47] presents evidence of the non-random structure of this network, which is related to the frequency of presented odors. Since the goal of our work is to build a useful AI system and not mimic every detail of the biological system, we adopt the data-driven synaptic weight strategy even if fruit flies may use random projections. As is clearly demonstrated in [39], learned synapses lead to better performance.

Our main contributions in this work are the following:

1. Inspired by the fruit fly network, we propose an algorithm that makes it possible to generate binary (as opposed to continuous) word embeddings for words and their context. We systematically evaluate the performance of this algorithm on word similarity task, word-sense disambiguation, and document classification.

2. We demonstrate that our binary embeddings result in tighter and better separated clusters of concepts compared to continuous GloVe embeddings, and stand in line with clustering properties of binarized versions of GloVe.

3. We show that training the fruit fly network requires an order of magnitude smaller compute time than training the classical NLP architectures, like BERT, at the expense of relatively small decrease in classification accuracy.

## 2    LEARNING ALGORITHM

Consider a training corpus. Each sentence can be decomposed into a collection of $w$-grams of consecutive words. If the word tokens come from a predefined vocabulary of size $N_{\text{voc}}$, the input to the algorithm is a vector of size $2N_{\text{voc}}$. This vector consists of two blocks: the context (the first $N_{\text{voc}}$ elements), and the target (the remaining $N_{\text{voc}}$ elements); see Fig. 2. In this work $w$ is assumed to be an odd integer, and the target word is assumed to be the center of the $w$-gram. The target word

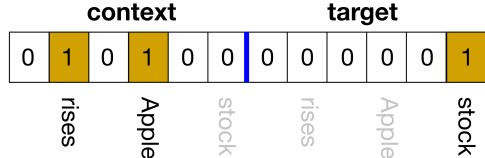

Figure 2: The encoding method. The input vector consists of two blocks separated by the (thick) blue line. Assuming $w = 3$, a center word "stock" is the target word and the two flanking words form a context. The $w$-gram is highlighted in light blue.

is one-hot encoded in the target block, and the context words are binary encoded as a bag of words in the context block (no positional information is used). The window $w$ slides along the text corpus, and for each position generates a training vector $\mathbf{v}^{\mathbf{A}} = \{v_i^A\}_{i=1}^{2N_{\text{voc}}}$, where the index $A$ enumerates different $w$-grams, and index $i$ enumerates positions in the context-target vector. These training vectors are passed to the learning algorithm. The goal of the algorithm is to learn correlations between the context and the target blocks.

### 2.1    MATHEMATICAL FORMULATION

Mathematically, the objective of the training algorithm is to distribute a set of context-target pairs among $K$ buckets, so that similar pairs end up in similar buckets. In order to achieve this, the learning algorithm takes two inputs: a set of training vectors $\mathbf{v}^{\mathbf{A}} \in \{0,1\}^{2N_{\text{voc}}}$, and a vector of occurrence probabilities $\mathbf{p} = \{p_i = f_{(i \mod N_{\text{voc}})}\}_{i=1}^{2N_{\text{voc}}} \in \mathbb{R}^{2N_{\text{voc}}}$, where $f_j$ is the probability of observing word $j$ in the training corpus[1]. The learning can be formalized as a minimization of the

---

[1]In our notation vector $\mathbf{f}$ has $N_{\text{voc}}$ elements, while vector $\mathbf{p}$ has $2N_{\text{voc}}$ elements. Given that index $i$ runs from 1 to $2N_{\text{voc}}$, notation $p_i = f_{(i \mod N_{\text{voc}})}$ is a mathematical way to concatenate two vectors $\mathbf{f}$ into a twice longer vector $\mathbf{p}$.

energy function, see [39] for additional details, defined by

$$E = - \sum_{A \in \text{data}} \frac{\left\langle \mathbf{W}_{\hat{\mu}}, \mathbf{v}^{\mathbf{A}}/\mathbf{p} \right\rangle}{\left\langle \mathbf{W}_{\hat{\mu}}, \mathbf{W}_{\hat{\mu}} \right\rangle^{1/2}}, \quad \text{where} \quad \hat{\mu} = \arg\max_{\mu} \left\langle \mathbf{W}_{\mu}, \mathbf{v}^{\mathbf{A}} \right\rangle \tag{1}$$

In this equation $\mathbf{W} \in \mathbb{R}^{K \times 2N_{\text{voc}}}$ is a matrix of synaptic connections, given as $\mathbf{W} = \{\mathbf{W}_{\mu}\} = \{W_{\mu i}\}$, projecting from PN layer (individual neurons in the layer are denoted by the index $i$) to the KC layer (individual neurons in the KC layer are denoted by the index $\mu$). There are $2N_{\text{voc}}$ neurons in the PN layer and $K$ neurons in the KC layer. The inner product $\langle \mathbf{X}, \mathbf{Y} \rangle = \sum_{i=1}^{2N_{\text{voc}}} X_i Y_i$ is defined as a contraction over index $i$ of PN cells. In the numerator of the energy function the binary encoded $w$-gram is divided by the probabilities of occurrences of individual words element-wise, so that the numerator can be written as

$$\left\langle \mathbf{W}_{\hat{\mu}}, \mathbf{v}^{\mathbf{A}}/\mathbf{p} \right\rangle = \sum_{i=1}^{2N_{\text{voc}}} W_{\hat{\mu} i} \frac{v_i^A}{p_i}$$

Probabilities $\mathbf{p}$ are calculated based on the frequencies of words in the training corpus. The vocabulary contains $N_{\text{voc}}$ most frequent words in the corpus, thus all the elements of $p_i$ are non-zero and the element-wise division is well defined.

Intuitively, the goal of the training algorithm is to adjust the weights of the neural network so that they are aligned with $w$-grams that are frequently present in the training corpus. We rely on the assumption that semantically related $w$-grams share several "core" words, while a few individual words might be substituted by synonyms/antonyms. The minimization of the energy function (1) is accomplished by the iterative update of the weights satisfying the following learning rule [25; 39; 17]

$$\Delta W_{\mu i} = \varepsilon g \Big[ \sum_j W_{\mu j} v_j^A \Big] \Big[ \frac{v_i^A}{p_i} - \Big( \sum_j W_{\mu j} \frac{v_j^A}{p_j} \Big) W_{\mu i} \Big] \tag{2}$$

In this equation the activation function is equal to one for a maximally driven hidden unit (Kenyon cell), and is equal to zero otherwise

$$g\Big[x_{\mu}\Big] = \delta_{\mu, \hat{\mu}}, \quad \text{where} \quad \hat{\mu} = \arg\max_{\mu} \Big[x_{\mu}\Big] \tag{3}$$

The learning rate is denoted by $\varepsilon$, and $\delta_{\mu, \hat{\mu}}$ is a Kronecker delta symbol.

## 2.2 Bio-Hashing

After learning is complete the hash codes for the inputs can be generated in the following way. Given the binary encoded $w$-gram $\mathbf{v}^{\mathbf{A}}$,

$$H_{\mu} = \begin{cases} 1, & \text{if } \langle \mathbf{W}_{\mu}, \mathbf{v}^{\mathbf{A}} \rangle \text{ in the top } k \text{ of all KCs activations} \\ 0, & \text{otherwise} \end{cases} \tag{4}$$

This is a crude mathematical approximation of the biological computation performed by the PN–KC–APL neural network [8; 39]. An input $\mathbf{v}^{\mathbf{A}}$ generates an input current $\langle \mathbf{W}_{\mu}, \mathbf{v}^{\mathbf{A}} \rangle$ into the KC neurons using feedforward weights $W_{\mu i}$. The recurrent network of KCs and the APL neuron silences all but a small fraction of KCs. Those cells that remain active are assigned state 1, while the rest of the KCs are assigned the inactive state 0.

Notice, that equation (4) makes it possible to generate the hash codes for both individual words (static word embeddings like word2vec and GloVe) and phrases (similar to Transformer models). In the static case, the input $\mathbf{v}^{\mathbf{A}}$ has all zeros in the context block and a one-hot encoded word in the target block. In the context-dependent case, both blocks have binary encoded input words. Importantly, both context-dependent and static embeddings are mapped into the same space of sparse binary hash codes (a vector of $K$ elements, with $k$ ones in it). We show below that these hash codes capture semantic meaning of the target word and the context in which it is used. For the rest of the paper we refer to the parameter $k$ in equation (4) as the *hash length*.

In order to provide an intuition behind the learning algorithm defined by the energy function (1) and weight update rule (2) and connect it to some of the existing methods in machine learning, consider

the limit when all the words have equal probabilities in the training corpus, $p_i = \frac{1}{N_{\text{voc}}}$. In this limit the energy function (1) reduces to the familiar spherical $K$-means clustering algorithm [11]. In this limit the weights of each KC correspond to the centroids of the clusters of context-target vectors. The hashing rule (4) assigns active state 1 to the $k$ closest centroids (and inactive state 0 to the remaining ones), defined with respect to cosine similarity distance. In this simple limit the learning algorithm that we use can be viewed as a biologically plausible implementation of this classical algorithm. For real datasets the probabilities of words are different, thus this correspondence does not hold. Notice that division by the probability appears only in the expression for the energy, but not in the definition of $\hat{\mu}$ in equation (1). Equivalently, division by $p_i$ appears in the second bracket of equation (2), but not in the argument of the activation function $g[x_\mu]$. Thus, in the general case (for different word probabilities $p_i$) our algorithm is not equivalent to spherical $K$-means on context-target vectors rescaled by the probabilities. Rather, in the general case, the closest centroid is found for a given context-target vector (via the definition of $\hat{\mu}$ in equation (1) - no $p_i$ involved), but the "updates of the position" of that centroid are computed by enhancing the contributions of rare words (small $p_i$) and suppressing the contributions of frequent words (large $p_i$). Empirically, we have found that division by the probabilities improves performance of our method compared to the case of spherical $K$-means (when the factor $1/\mathbf{p}$ is removed from the algorithm).

## 3 EMPIRICAL EVALUATION

The KC network shown in Fig. 1 was trained on the OpenWebText Corpus [15], which is a 32GB corpus of unstructured text containing approximately 6B tokens. The details of the training protocols and the hyperparameters are reported in section 7 in the supplement.

### 3.1 STATIC WORD EMBEDDINGS EVALUATION

Our aim here is to demonstrate that the sparse embeddings obtained by the fruit fly network motif are competitive with existing state-of-the-art word embeddings such as GloVe [34] and word2vec [30] and commonly used binarization tools for these continuous embeddings. We show this by evaluating the semantic similarity of static word embeddings. Several common benchmark datasets are used: WS353 [13], MEN [4], RW [28], SimLex [21], RG-65 [38], Mturk [18]. These datasets contain pairs of words with human-annotated similarity scores between them. Following previous work [43; 42], model similarity score for binary representations is evaluated as $sim(v_1, v_2) = (n_{11} + n_{00})/n$, where $n_{11}$ ($n_{00}$) is the number of bits in $v_1$ and $v_2$ that are both 1 (0), and $n$ is the length of $v_{1,2}$. Cosine similarity is used for real-valued representations. Spearman's correlation coefficient is calculated between this similarity and the human annotated score. The results are reported in Table 1.

| Dataset | Ours | GloVe | word2vec | SOTA | |
|---------|------|-------|----------|------|------|
| MEN | 56.6 | 69.5 | 75.5 | 81.3 | [12] |
| WS353 | 63.7 | 64.0 | 66.5 | 81.0 | [18] |
| SIMLEX | 21.0 | 31.5 | 41.7 | 56.0 | [40] |
| RW | 39.4 | 46.8 | 61.3 | 61.7 | [36] |
| RG | 69.0 | 74.2 | 75.4 | 83.3 | [20] |
| Mturk | 56.1 | 57.5 | 69.8 | 72.7 | [18] |

Table 1: Evaluation on word similarity datasets via Spearman's rank correlation coefficient. Both GloVe and word2vec use 300d pretrained embeddings. Hyperparameter settings for our model: $K = 400$, $w = 11$. Results for our algorithm are reported only for a fixed hash length, $k = 51$. See Table 7 for results as a function of hash length.

We observe that our word embeddings demonstrate competitive performance compared to GloVe, but worse performance than word2vec. At the same time, our embeddings are binary, as opposed to GloVe and word2vec, which are represented by continuous vectors. Thus, it is more appropriate to compare them with commonly used binarized versions of the continuous embeddings. Specifically, we compare the performance of fruit fly embeddings with a number of state-of-the-art binarization methods such as: LSH/SimHash [7] (random *contractive* projections followed by binarization based on sign), RandExp [8] (random *expansive* projections followed by $k$-winner take all binarization),

ITQ [16] (iterative quantization), SH (spectral hashing) [46], PCAH [16] (PCA followed by binarization based on sign). The complete evaluation of all these methods for varying hash length is presented in Section 8; please see Tables 7, 8, 9 for binarization of pretrained GloVe, pretrained word2vec, and GloVe trained on OpenWebText. In Table 7 we also include evaluation from NLB, "Near-Lossless Binarization" [43] (autoencoder-based binarization) for the hash lengths where those results are available. Here we only present a short summary of those results for a specific (small) hash length $k = 4$ in Table 2.

| Dataset | Ours | LSH | RandExp | ITQ | SH | PCAH |
|---------|------|-----|---------|-----|-----|------|
| MEN | 34.0 | 16.9/**35.5**/23.6 | 27.5/24.2/28.4 | 0.1/9.2/26.9 | 9.4/7.2/23.8 | 12.5/5.3/26.0 |
| WS353 | **43.2** | 8.2/26.0/20.2 | 20.9/23.5/30.5 | -6.6/16.0/25.9 | 15.4/3.3/18.1 | 6.4/17.3/21.2 |
| SIMLEX | 13.4 | 6.8/17.0/8.0 | 10.4/**17.6**/10.1 | 7.0/3.3/7.3 | 9.3/-3.6/12.1 | 4.4/-2.9/11.5 |
| RW | 11.0 | 10.8/21.8/16.2 | 19.9/**24.7**/22.0 | 13.7/17.4/24.5 | 22.6/14.6/19.7 | 12.4/15.0/19.7 |
| RG | 24.0 | 21.2/44.6/25.5 | 36.6/30.4/28.7 | -17.5/32.8/21.4 | 4.5/18.0/39.8 | 1.9/20.8/**45.0** |
| Mturk | **44.0** | 16.0/33.1/18.3 | 29.3/22.7/28.3 | 9.9/22.5/26.3 | 18.9/21.9/20.3 | 15.5/23.6/24.9 |

Table 2: Comparison to common binarization methods. This table is a simplified version (for hash length $k = 4$) of the complete evaluation for a range of hash lengths reported in Tables 7, 8, 9. Each binarization technique was evaluated on three continuous embeddings: pretrained GloVe, pretrained word2vec, GloVe trained on OpenWebText (the same dataset that was used for training our fruit fly embeddings), format: pretrained GloVe/ pretrained word2vec/ GloVe on OWT. Hyperparameter settings for our model: $K = 400$, $w = 11$. Best result in bold; second best underlined.

It is clear from Table 2 that fruit fly embeddings outperform existing methods for word embedding discretization on WS353 and Mturk, and demonstrate second best result (after LSH binarization of word2vec) on MEN. In general (see Tables 7, 8, 9), we find that fruit fly embeddings are particularly powerful compared to existing methods at small hash lengths (see $k = 4, 8$ in the aforementioned tables). These results indicate that the fruit fly network can learn meaningful binary semantic representations directly from raw text. We also note that an added advantage of binary embeddings is that they require only a fraction (approx. 3%) of the memory footprint required for continuous word embeddings (assuming they have the same length), since a real value requires 32-bits per vector element, whereas a boolean value requires only 1-bit.

## 3.2 WORD CLUSTERING

A nice aspect of binary embeddings is that they result in tighter and better separated clusters than continuous embeddings. To evaluate this property for our method we started with hash codes for individual words and performed agglomerative clustering via complete link, using the cosine distance as the metric. The clustering algorithm was terminated at 200 clusters (we experimented with possible choices of this parameter, such as $200, 500, 1000, 2000, 3000, 5000$, and arrived at similar conclusions). We repeated the same analysis for continuous GloVe, binarization of GloVe embeddings via autoencoder-like method [43], and simple discretization method of GloVe when one declares the largest $k$ elements of each word vector to be 1 and assigns 0 to the remaining elements (for $k = 50, 75, 120, 200$). The results for the inter-cluster similarity vs. intra-cluster similarity are shown in Fig. 3 (panel A). It is clear from this scatter plot that the average distance between the points within a cluster is smaller (higher similarity) for all considered binary embeddings compared to GloVe embeddings. At the same time, the distance between the closest clusters is larger or equal (smaller similarity) for the fruit fly embeddings and naive discretizations with $k < \approx 120$. We also observe that the clusters lose detail (i.e., both intra- and inter-cluster similarity increases) as the binarization threshold gets higher (shown for Glove). However, our embeddings maintain a balance between intra- and inter-clustering similarity, and thus still capture fine-grained cluster information. For instance, inspecting the semantic structure of the clusters obtained this way, an example of the hierarchical clustering diagram (lower part of the tree containing 42 leaves) is shown in Fig. 3 (panel B). We clearly observe semantically coherent clusters resulting from the fruit fly word embeddings.

## 3.3 CONTEXT-DEPENDENT WORD EMBEDDINGS

Here, we evaluate the effectiveness of our fruit fly inspired approach for contextual word embeddings, as opposed to static (or context-independent) embeddings from above. We use the WiC [35]

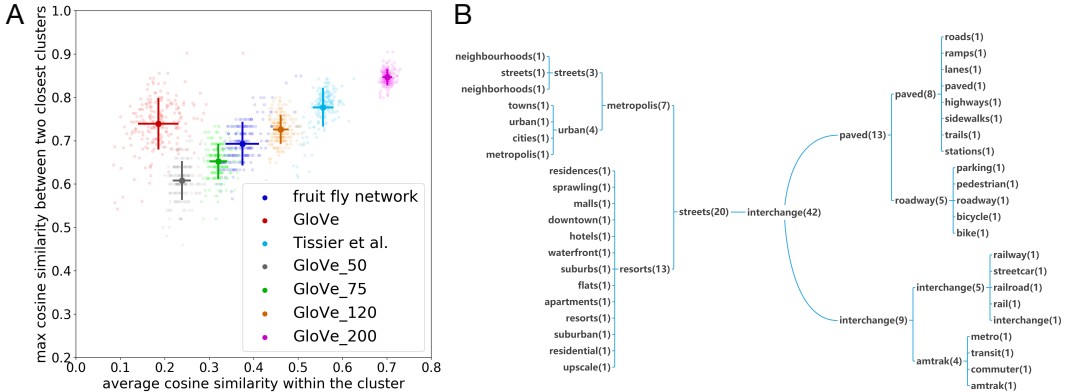

Figure 3: Panel A: average cosine similarity between the points within the cluster vs. maximum cosine similarity (minimal distance) to a point from the closest cluster. Solid lines correspond to mean±std for the individual clusters. Numbers next to GloVe in the legend correspond to the number of largest elements in the word vector that are mapped to 0 under the naive discretization procedure. Panel B: an example of a cluster generated by the agglomerative clustering for our method, the integer number associated with each node corresponds to the number of daughter leaves in that cluster. The root node corresponds to "interchange (42)".

and SCWS [22] benchmarks for the evaluation of context-sensitive word embeddings for word sense disambiguation. Both the datasets comprise pairs of sentences that contain a target word, and the task is to determine whether the two target words share a similar semantic meaning in the corresponding contexts. The WiC dataset is modeled as a binary prediction task, with 1 denoting that the target words have the same sense, and 0 indicating that they mean different things. The SCWS dataset is modeled as a rank prediction task, since for each pair of sentences and target words, it reports the average human similarity scores (from 10 Amazon Mechanical Turkers per pair).

| word in context | 10 nearest neighbor words in the hash code space |
|---|---|
| design of **apple** latest iphone | design, web, features, graphics, radeon, android, apps, ios, apple, nvidia |
| filling sweet **apple** pie recipe | chocolate, sweet, crispy, noodles, syrup, coconut, cheese, sauce, butter, cinnamon |
| money in **bank** checking account | money, credit, loans, account, cash, funds, savings, paying, banks, pension |
| boat on the **bank** of the river | river, lake, near, island, creek, canyon, valley, mountains, area, shore |

Figure 4: For every word (highlighted in green) in context (left), 10 nearest neighbor words in the binary hashing space are shown (right). Context allows the algorithm to disambiguate the target word's meaning.

Before presenting quantitative results, we qualitatively examine how the fruit fly network performs on context sentence pairs for target words "apple" and "bank" in Fig. 4. We show the top $q = 10$ nearest neighbor words for the context dependent target word. These examples clearly indicate that the "correct" sense of the word has been found ("apple" the device manufacturer has different nearest neighbors from the fruit, and "bank" the financial institution from the natural feature).

For the quantitative comparison, we contrast our method against contextual embeddings from BERT [10], GloVe [34], word2vec [30] and Word2Sense [33]. For BERT we use the 768-dimensional embeddings from the uncased-large model, for GloVe and word2vec we use the 300-dimensional embeddings, and for Word2Sense we use the sparse 2250-dimensional pretrained embeddings. Since BERT outputs contextual embeddings for each word in a sentence, we simply compute the cosine similarity between the embedding vectors for the target words for each pair of instances. For GloVe/word2vec, we use a context window of size $w$ centered at each of the target words and compute the average embedding for each window and compute the cosine similarity between the two window vectors. Similar approach is used for Word2Sense, but the similarity between two embeddings is based on the Jensen-Shannon divergence [33]. For the fruit fly network, given the effectiveness of the top-$q$ nearest neighbor words (as seen in Fig. 4), we devise a two component scoring function. The first component is the dot-product between the context-dependent hash codes for the two target words plus $w$ length context blocks, denoted $J_{dot}$. The second is the number of

common contextual nearest neighbors of the two target words among the top-$q$ neighbors of each (scaled to be between 0 and 1), denoted $J_{nn}$. The final score is given as $J = \alpha \cdot J_{dot} + (1 - \alpha) \cdot J_{nn}$, where $\alpha \in [0, 1]$ is a hyperparameter. For all the methods, we predict a WiC pair to be positive if the score is above a threshold value $\theta$. For SCWS, the ranking is proportional to the scores above $\theta$, with the rest scored as zero. The hyperparameter $\theta$ is tuned for all the methods independently. Finally, for a fair comparison, all methods use the same 20k vocabulary.

We report the performance of our context-dependent word embeddings for both SCWS and WiC in Table 3 and Table 4, respectively. For both benchmarks we report the results from a 5-fold cross-validation study, where each fold (in turn) is used as a development set, and the remaining four folds as the test set. We select the optimal hyperparameters (including $\theta, \alpha, q, k, w$) for all the methods using only the first fold; no training is done since we evaluate only the pretrained embeddings. The tables report the Spearman rank correlation on SCWS, and the accuracy on WiC.

| Method | mean | std |
|---|---|---|
| BERT | **56.8** | 0.54 |
| word2vec ($w = 0$) | 56.7 | 0.005 |
| GloVe ($w = 3$) | 40.9 | 1.3 |
| GloVe ($w = 0$) | 54.4 | 0.10 |
| Word2Sense ($w = 3$) | 41.4 | 0.01 |
| Word2Sense ($w = 0$) | 54.2 | 0.008 |
| Ours ($w = 0$) | 49.1 | 0.36 |

Table 3: SCWS dataset: mean and std for Spearman rank correlation. The best window value is also shown.

| Method | mean | std |
|---|---|---|
| BERT | **61.2** | 0.22 |
| word2vec ($w = 5$) | 51.3 | 0.004 |
| Word2vec ($w = 0$) | 50.0 | 0.003 |
| GloVe ($w = 7$) | 54.9 | 0.26 |
| GloVe ($w = 0$) | 50.1 | 0.25 |
| Word2Sense ($w = 7$) | 56.5 | 0.004 |
| Word2Sense ($w = 0$) | 50.0 | 0.003 |
| Ours ($w = 21$) | 57.7 | 0.27 |

Table 4: WiC dataset: mean and std for accuracy. The best window value is also shown.

On SWCS (Table 3), we see that the context-independent embeddings (using $w = 0$) are better for GloVe, Word2Sense and our method, with word2vec yielding the best results. The reason is that about 86.5% of the word pairs in SCWS are different words, and can be distinguished without looking at the context. Unlike SCWS, the WiC benchmark uses the same target word (with only minor variations in some cases) in both contexts, and therefore a context-independent approach is not expected to perform well. Indeed, on WiC (Table 4), we clearly observe that context-independent vectors ($w = 0$) are not very good, and our method, that uses the joint scoring function $J$ combining both the hash code and nearest neighbor scores, is better than context-dependent GloVe ($w = 7$), word2vec ($w = 5$) and Word2Sense (also $w = 7$).

| Dataset | Ours | Glove | NLB(256bits) | NLB(512bits) | Word2vec | BERT |
|---|---|---|---|---|---|---|
| 20Newsgroup | 78.2 | 77.9 | 61.6 | 64.1 | 77.3 | **78.6** |
| SST-2 | 77.1 | 78.3 | 76.3 | 78.6 | 80.7 | **90.8** |
| WOS-11967 | 83.8 | 84.2 | 70.6 | 72.8 | 84.8 | **86.7** |
| TREC-6 | 90.4 | 89.0 | 85.2 | 88.8 | 90.9 | **94.0** |

Table 5: Accuracy for document classification task. We use 300d pretrained models for GloVe and word2vec, and pretrained bert-large-uncased model for BERT. For NLB, 300d GloVe embeddings were binarized into 256 and 512 bits. For our model, hash length 30 is used. For fair comparison, all models use the same vocabulary of 20k words.

## 3.4 DOCUMENT CLASSIFICATION

We also compare our binary embeddings with GloVe [34], Word2vec [31], BERT [10] and Near-Lossless Binarization [43] on document classification tasks. The benchmarks we use are 20 Newsgroups [1], Stanford Sentiment Treebank [41], WOS-11967[24] and TREC-6 datasets [27]. The 20 Newsgroups dataset contains around 18,000 documents, partitioned evenly into 20 different groups; the Stanford Sentiment Treebank dataset contains movie reviews reflecting their sentiment as either positive or negative; WOS-11967 dataset contains 11967 documents with 35 categories which include 7 parents categories; and TREC-6 dataset consists of open-domain, fact-based questions divided into broad semantic categories. We use the TextCNN [23] classifier that uses all the different embeddings mentioned above. For fair comparison, we use the same model parameters (e.g., kernel size, filter dimension) while testing different embeddings. The results in Table 5 show how our sparse binary encodings are competitive with other methods.

| device | $K$ | batch-size | GPU mem | time |
|--------|-----|-----------|---------|------|
| V100 × 3 | 400 | 2000× 3 | 122MB | 17m |
| V100 × 3 | 400 | 10000× 3 | 150MB | 8m |
| V100 × 3 | 600 | 2000× 3 | 232MB | 24m |
| V100 × 3 | 600 | 10000*3 | 267MB | 11.5m |
| CPU 44cores | 400 | 2000 | - | 76m |
| CPU 44cores | 400 | 10000 | - | 25m |

Table 6: Training time (per epoch) and memory footprint of our method on GPUs and CPUs. For the GPU implementation, three V100 GPUs interconnected with 100GB/s (bidirectional) NVLink were used. For the CPU implementation, the computation was done on two 22-core CPUs. CPU memory is 137GB. The results are reported for window $w = 11$.

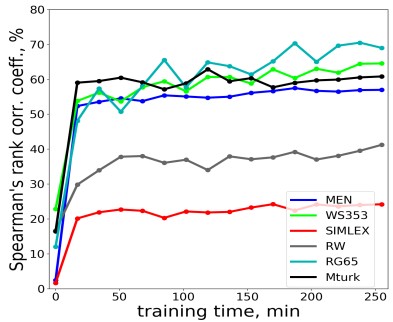

Figure 5: Spearman's correlation on word similarity datasets (see Section 3.1) vs. training time. Each point is one epoch.

## 4 COMPUTATIONAL COMPLEXITY

The computational complexity of our method can be evaluated by analyzing equations (2,3) for the weight updates. In these equations $\mathbf{v}^{\mathbf{A}}$ is a sparse vector, which has only $w$ non-zero elements in it. Thus, for a minibatch of size $|BS|$, the computational complexity of evaluating the dot product with weights is $K \cdot w \cdot |BS|$. Additionally, the argmax operation requires $K \cdot |BS|$ operations. We will assume that the largest parameters in our model are the size of the corpus $|A| \approx 10^{10}$, and the size of the vocabulary $N_{\text{voc}} \approx 10^4 - 10^5$. Additionally we use large minibatches $|BS| \approx N_{\text{voc}}$. Calculation of the second term in (2) requires $O(K \cdot N_{\text{voc}})$ operations in addition to $K \cdot w \cdot |BS|$ operations for calculating the dot-product for each data point. Since the algorithm has to go over the entire corpus, this computation needs to be repeated $|A|/|BS|$ times per epoch. Thus, the overall computational complexity of our method is $O\left(K \cdot |A|(w + N_{\text{voc}}/|BS|)\right) \approx K \cdot |A| \cdot w$ per epoch. Thus, in the leading order it does not grow with the size of the vocabulary, which is a nice computational feature.

From the practical perspective, typical wall-clock training time and memory requirements per epoch are shown in Table 6. As is shown in Fig. 5, accurate solutions are obtained after about $2 - 3$ epochs; improvements beyond that are relatively small. Thus, our algorithm is capable of producing competitive models in a couple of hours. Contrast this with approximately 24 hours training time for GloVe [34]; 4 days of training on 16 TPUs for BERT$_{\text{BASE}}$; and 4 days on 64 TPUs for BERT$_{\text{LARGE}}$ [10] (the last two numbers assume training corpus of size 250B tokens vs. 6B tokens considered in this paper). The record breaking training time of 47 minutes for BERT requires the use of 1472 NVIDIA V100 GPUs each with 32GB of memory and a specialized DGX server architecture [32]. In our own experiments, we trained GloVe embedding on OWT corpus using the same vocabulary of 20k words that we used for the fruit fly embeddings. The wall-clock training time was approximately 10 hours on 16 threads, see details in Section 11. These are substantially larger computational resources than those required for training the fruit fly network.

## 5 DISCUSSION AND CONCLUSIONS

In this work we asked the intriguing question whether the core computational algorithm of one of the best studied networks in neuroscience – the network of KCs in the fruit fly brain – can be repurposed for solving a well defined machine learning task, namely, learning word embeddings from text. We have shown that, surprisingly, this network can indeed learn the correlations between the words and their context, and produce high quality word embeddings. On the semantic similarity task the fruit fly word embeddings outperform common methods for binarizing continuous SOTA word embeddings (applied to GloVe, word2vec, and GloVe trained on OWT) at small hash lengths. On the word-in-context task the fruit fly network outperforms GloVe by almost $3\%$, word2vec by more than $6\%$, but looses to BERT by $3.5\%$, see Table 4. The small gap in classification accuracy compared with BERT, however, is outweighed by the benefit of requiring significantly smaller computational resources to obtain these fruit fly embeddings, as we have explained in Section 4, see Table 6. We view this result as an example of a general statement that biologically inspired algorithms might be more compute efficient compared with their classical (non-biological) counterparts, even if they slightly lose in terms of accuracy.

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

# 6    APPENDIX A. RELATED WORK.

Our work builds on several ideas previously discussed in the literature. The first idea is that fruit fly olfactory network can generate high quality hash codes for the input data in random [8] and data-driven [39] cases. There are two algorithmic differences of our approach compared to these previous studies. First, our network uses representational contraction, rather than expansion when we go from the PN layer to the KCs layer. Second, [8; 39] construct hash codes for data coming from a single modality (e.g., images, or word vectors), while the goal of the present paper is to learn correlations between two different "modalities": target word and its context. The second idea pertains to the training algorithm for learning the PN→KCs synapses. We use a biologically plausible algorithm of [25] to do this, with modifications that take into account the wide range of frequencies of different words in the training corpus (we discuss these differences in section 2.1). Also, similarly to [8; 39] the algorithm of [25] is used for learning the representations of the data, and not correlations between two types of data (context and target) as we do in this paper.

Another closely related work [37] uses the network of KCs with random weights for generating binary hash codes for individual words. There are several differences compared to our approach. First, in our system the synaptic weights from PNs to KCs are learned and not random. We have found that learning these weights improves the performance compared to the random case. Second, unlike [37] (and unlike fruit flies), in our system the number of KCs is smaller than the number of PNs, so there is no representational expansion as we move into the "mushroom body". This expansion is essential for the system of [37], which uses random weights. Finally, our algorithm uses a different encoding scheme at the level of PNs, see Fig. 2.

# 7    APPENDIX B. TRAINING PROTOCOLS AND HYPERPARAMETER CHOICES.

The fruit fly network was trained on the OpenWebText Corpus [15], which is a 32GB corpus of unstructured text containing approximately 6B tokens. Individual documents were concatenated and split into sentences. A collection of $w$-grams were extracted from each sentence by sliding a window of size $w$ along each sentence from the beginning to the end. Sentences shorter than $w$ were removed. The vocabulary was composed of $N_{\text{voc}} = 20000$ most frequent tokens in the corpus.

Training was done for $N_{\text{epoch}}$. At each epoch all the $w$-grams were shuffled, organized in minibatches, and presented to the learning algorithm. The learning rate was linearly annealed starting from the maximal value $\varepsilon_0$ at the first epoch to nearly zero at the last epoch.

The training algorithm has the following hyperparameters: size of the KC layer $K$, window $w$, overall number of training epochs $N_{\text{epoch}}$, initial learning rate $\varepsilon_0$, minibatch size, and hash length $k$. All models presented in this paper were trained for $N_{\text{epoch}} = 15$. The optimal ranges of the hyperparameters are: learning rate is $\varepsilon_0 \approx 10^{-4} - 5 \cdot 10^{-4}$; $K \approx 200 - 600$; $w \approx 9 - 15$; minibatch size $\approx 2000 - 15000$; hash length $k$ is reported for each individual experiment.

# 8    APPENDIX C. COMPARISON WITH BINARIZED GLOVE AND WORD2VEC.

Our aim here is to demonstrate that the fruit fly word embeddings are competitive with existing state-of-the-art binarization methods applied to GloVe and word2vec embeddings. We show this by evaluating the semantic similarity of static word embeddings, using several common benchmark datasets: WS353 [13], MEN [4], RW [28], SimLex [21], RG-65 [38], and Mturk [18]. These datasets contain pairs of words with human-annotated similarity scores between them. Specifically, we compare with GloVe [34] word embeddings[2] trained on Wiki2014 and Gigaword 5, GloVe embeddings trained on OpenWebText Corpus [15] and word2vec embeddings[3].

Since our representations are binary (in contrast to GloVe and word2vec), we binarize GloVe and word2vec embeddings and report their performance using a number of common hashing methods, LSH/SimHash [7] (random *contractive* projections followed by binarization based on sign), RandExp [8] (random *expansive* projections followed by $k$-winner take all binarization), ITQ [16] (it-

---

[2]pretrained embeddings: https://nlp.stanford.edu/projects/glove
[3]pretrained embeddings: https://code.google.com/archive/p/word2vec

| Method | Hash Length ($k$) | | | | | | Hash Length ($k$) | | | | | |
|---|---|---|---|---|---|---|---|---|---|---|---|---|
| | 4 | 8 | 16 | 32 | 64 | 128 | 4 | 8 | 16 | 32 | 64 | 128 |
| | **MEN** (69.5/68.1) | | | | | | **WS353** (64.0/47.7) | | | | | |
| Ours | **34.0** | **49.9** | **55.9** | 56.7 | 55.3 | 51.3 | **43.2** | **52.1** | 55.3 | 57.4 | **60.3** | 51.7 |
| LSH | 16.9 | 23.7 | 35.6 | 42.6 | 53.6 | 63.4 | 8.2 | 20.7 | 30.0 | 34.7 | 43.9 | 50.3 |
| RandExp | 27.5 | 37.7 | 46.6 | **57.6** | **67.3** | **71.6** | 20.9 | 32.9 | 41.9 | 48.4 | 57.6 | 61.7 |
| ITQ | 0.1 | 7.7 | 10.5 | 16.5 | 30.4 | 50.5 | -6.6 | -6.1 | -2.4 | -4.4 | 6.1 | 24.8 |
| SH | 9.4 | 17.0 | 22.9 | 37.6 | 52.9 | 65.4 | 15.4 | 14.1 | 19.5 | 32.3 | 43.1 | 58.4 |
| PCAH | 12.5 | 21.8 | 27.6 | 39.6 | 53.4 | 68.1 | 6.4 | 6.3 | 20.6 | 33.9 | 49.8 | **62.6** |
| NLB | - | - | - | - | 46.1 | 63.3 | - | - | - | - | 30.1 | 44.9 |
| | **SIMLEX** (31.5/29.8) | | | | | | **RW** (46.8/31.4) | | | | | |
| Ours | **13.4** | 16.5 | **22.8** | 22.1 | 21.1 | 17.0 | 11.0 | **22.6** | 25.8 | 36.9 | 38.6 | 35.2 |
| LSH | 6.8 | 11.9 | 17.0 | 21.2 | 26.8 | 30.9 | 10.8 | 16.3 | 21.8 | 27.8 | 36.3 | 45.0 |
| RandExp | 10.4 | **17.2** | **22.8** | **28.5** | **32.4** | **35.2** | 19.9 | 21.3 | **30.9** | **40.5** | **47.6** | **53.3** |
| ITQ | 7.0 | 1.6 | 4.3 | 5.5 | 11.8 | 18.2 | 13.7 | 5.3 | 6.6 | 6.9 | 12.5 | 26.5 |
| SH | 9.3 | 15.6 | 15.9 | 17.0 | 23.1 | 31.2 | **22.6** | 21.5 | 24.3 | 28.8 | 36.1 | 45.8 |
| PCAH | 4.4 | 10.3 | 11.0 | 17.3 | 24.1 | 31.6 | 12.4 | 16.7 | 21.5 | 30.3 | 36.9 | 44.4 |
| NLB | - | - | - | - | 20.5 | 31.4 | - | - | - | - | 25.1 | 34.3 |
| | **RG** (74.2/67.6) | | | | | | **Mturk** (57.5/61.9) | | | | | |
| Ours | 24.0 | 40.4 | **51.3** | 62.3 | 63.2 | 55.8 | **44.0** | **49.0** | **52.2** | **60.1** | 57.7 | 55.2 |
| LSH | 21.2 | 35.4 | 44.6 | 55.1 | 63.1 | 70.1 | 16.0 | 23.1 | 33.2 | 35.6 | 42.7 | 55.5 |
| RandExp | **36.6** | **49.0** | 49.5 | **66.1** | **69.6** | 70.9 | 29.3 | 35.8 | 41.4 | 50.4 | **59.0** | **61.6** |
| ITQ | -17.5 | -8.9 | 26.3 | 41.7 | 50.5 | 66.2 | 9.9 | 7.8 | 10.1 | 17.7 | 32.8 | 47.3 |
| SH | 4.5 | 5.8 | 20.3 | 42.9 | 61.3 | **72.6** | 18.9 | 17.6 | 27.5 | 35.45 | 48.1 | 57.9 |
| PCAH | 1.9 | 9.6 | 19.8 | 40.9 | 53.3 | 68.2 | 15.5 | 15.1 | 27.1 | 41.7 | 46.5 | 56.2 |

Table 7: Evaluation on word similarity datasets. For each dataset and hash length, the best (second best) score is in **bold** (underlined). The performance for GloVe embeddings is reported next to the name of each dataset in the format 300d/100d. Spearman's rank correlation coefficient is reported for common baselines that binarize GloVe (300d) embeddings together with our results. Hyperparameter settings for our algorithm: $K = 400$, $w = 11$.

erative quantization), SH (spectral hashing) [46], PCAH [16] (PCA followed by binarization based on sign). Where available, we include evaluation from NLB, "Near-Lossless Binarization" [43] (autoencoder-based binarization).

Following previous work [43; 42], model similarity score for binary representations is evaluated as $sim(v_1, v_2) = (n_{11} + n_{00})/n$, where $n_{11}$ ($n_{00}$) is the number of bits in $v_1$ and $v_2$ that are both 1 (0), and $n$ is the length of $v_{1,2}$. Cosine similarity is used for real-valued representations. The results are reported in Tables 7, 8 and 9. For each dataset, we report performance across a range of hash lengths $\{4, 8, 16, 32, 64, 128\}$. For methods that incorporate randomness (LSH, RandExp, ITQ), we report the average across 5 runs. ITQ, SH and PCAH in Tables 7 and 8 were trained using the top 400k most frequent words. Table 9 compares our method to GloVe trained on OpenWebText (same dataset that our method is trained on) using the same vocabulary as our method uses.

Our binary word embeddings demonstrate competitive performance compared to published methods for GloVe and word2vec binarization, and our algorithm can learn meaningful binary semantic representations directly from raw text. Importantly, our algorithm does not require training GloVe or word2vec embeddings first before binarizing them.

## 9 APPENDIX D. DETAILS OF TECHNICAL IMPLEMENTATION.

From the practical perspective, efficient implementation of the learning algorithm for the fruit fly network requires the use of sparse algebra, atomic updates, and block-sparse data access. Our algorithm is implemented in CUDA as a back-end, while python is used as an interface with the main functions.

The typical memory footprint of our approach is very small. About $100 - 270$MB GPU memory is allocated for the operators $W_{\mu i}$, $\mathbf{v}^{\mathbf{A}}$ and temporary fields; while approximately 140GB CPU memory is needed to store the input data, array of random numbers for shuffle operations and shuffled indices. For GPU implementation, the model data is stored in the GPU's memory, while the input data

| Method | Hash Length ($k$) | | | | | | Hash Length ($k$) | | | | | |
|---|---|---|---|---|---|---|---|---|---|---|---|---|
| | 4 | 8 | 16 | 32 | 64 | 128 | 4 | 8 | 16 | 32 | 64 | 128 |
| | **MEN** (75.5) | | | | | | **WS353** (66.5) | | | | | |
| Ours | 34.0 | 49.9 | 55.9 | 56.7 | 55.3 | 51.3 | 43.2 | 52.1 | 55.3 | 57.4 | 60.3 | 51.7 |
| LSH | 35.5 | 42.5 | 53.6 | 63.4 | 68.4 | 72.2 | 26.0 | 34.7 | 43.9 | 50.3 | 56.0 | 58.6 |
| RandExp | 24.2 | 34.6 | 45.8 | 57.5 | 66.1 | 71.7 | 23.5 | 34.3 | 37.3 | 48.0 | 57.6 | 63.7 |
| ITQ | 9.2 | 13.3 | 25.1 | 41.5 | 57.6 | 68.5 | 16.0 | 18.1 | 22.5 | 30.2 | 43.9 | 54.8 |
| SH | 7.2 | 15.8 | 31.3 | 46.9 | 62.3 | 69.4 | 3.3 | 9.6 | 22.7 | 34.1 | 50.0 | 54.7 |
| PCAH | 5.3 | 18.6 | 37.7 | 52.0 | 63.9 | 71.6 | 17.3 | 24.9 | 38.5 | 42.0 | 52.1 | 59.3 |
| | **SIMLEX** (41.7) | | | | | | **RW** (61.3) | | | | | |
| Ours | 13.4 | 16.5 | 22.8 | 22.1 | 21.1 | 17.0 | 11.0 | 22.6 | 25.8 | 36.9 | 38.6 | 35.2 |
| LSH | 17.0 | 21.2 | 26.8 | 30.9 | 34.4 | 35.1 | 21.8 | 27.8 | 36.3 | 45.0 | 49.6 | 52.1 |
| RandExp | 17.6 | 24.4 | 29.2 | 32.6 | 38.0 | 39.8 | 24.7 | 27.7 | 39.8 | 46.8 | 52.3 | 55.6 |
| ITQ | 3.25 | 5.7 | 6.2 | 14.9 | 23.1 | 31.5 | 17.4 | 15.7 | 19.1 | 33.5 | 45.6 | 53.4 |
| SH | -3.6 | 3.6 | 10.4 | 17.0 | 23.7 | 32.4 | 14.6 | 22.8 | 28.7 | 37.9 | 43.5 | 52.4 |
| PCAH | -2.9 | 2.5 | 11.8 | 17.0 | 24.0 | 36.0 | 15.0 | 21.5 | 28.8 | 35.4 | 46.4 | 50.6 |
| | **RG** (75.4) | | | | | | **Mturk** (69.8) | | | | | |
| Ours | 24.0 | 40.4 | 51.3 | 62.3 | 63.2 | 55.8 | 44.0 | 49.0 | 52.2 | 60.1 | 57.7 | 55.2 |
| LSH | 44.6 | 55.1 | 63.1 | 70.1 | 76.4 | 75.8 | 33.1 | 35.6 | 42.7 | 55.5 | 58.6 | 62.4 |
| RandExp | 30.4 | 42.0 | 48.6 | 59.1 | 70.2 | 74.6 | 22.7 | 34.8 | 42.0 | 45.9 | 57.9 | 61.2 |
| ITQ | 32.8 | 49.7 | 31.5 | 55.9 | 62.2 | 71.6 | 22.5 | 21.3 | 42.3 | 46.9 | 59.3 | 60.7 |
| SH | 18.0 | 30.6 | 36.0 | 48.8 | 56.9 | 75.8 | 21.9 | 27.4 | 41.8 | 51.2 | 58.8 | 58.0 |
| PCAH | 20.8 | 22.9 | 40.6 | 36.5 | 59.0 | 71.2 | 23.6 | 34.4 | 45.5 | 55.7 | 64.2 | 60.5 |

Table 8: Evaluation on word similarity datasets, analogous to Table 7, for 300d word2vec embeddings.

| Method | Hash Length ($k$) | | | | | | Hash Length ($k$) | | | | | |
|---|---|---|---|---|---|---|---|---|---|---|---|---|
| | 4 | 8 | 16 | 32 | 64 | 128 | 4 | 8 | 16 | 32 | 64 | 128 |
| | **MEN** (76.4) | | | | | | **WS353** (72.2) | | | | | |
| Ours | 34.0 | 49.9 | 55.9 | 56.7 | 55.3 | 51.3 | 43.2 | 52.1 | 55.3 | 57.4 | 60.3 | 51.7 |
| LSH | 23.6 | 29.1 | 37.4 | 49.6 | 60.6 | 67.0 | 20.2 | 29.0 | 35.5 | 47.5 | 53.3 | 61.4 |
| RandExp | 28.4 | 40.3 | 52.3 | 62.5 | 67.7 | 71.0 | 30.5 | 40.0 | 48.1 | 57.9 | 63.3 | 67.5 |
| ITQ | 26.9 | 33.9 | 46.3 | 56.1 | 64.1 | 70.3 | 25.9 | 33.7 | 44.5 | 56.1 | 63.9 | 67.6 |
| SH | 23.8 | 28.7 | 44.1 | 54.7 | 62.1 | 69.7 | 18.1 | 25.7 | 40.1 | 51.8 | 60.9 | 62.9 |
| PCAH | 26.0 | 30.1 | 46.3 | 57.9 | 67.5 | 72.4 | 21.2 | 30.5 | 43.8 | 50.7 | 61.1 | 59.9 |
| | **SIMLEX** (34.0) | | | | | | **RW** (54.5) | | | | | |
| Ours | 13.4 | 16.5 | 22.8 | 22.1 | 21.1 | 17.0 | 11.0 | 22.6 | 25.8 | 36.9 | 38.6 | 35.2 |
| LSH | 8.0 | 16.8 | 19.0 | 24.8 | 26.7 | 32.9 | 16.2 | 21.0 | 26.1 | 33.6 | 40.8 | 47.0 |
| RandExp | 10.1 | 17.3 | 23.4 | 26.6 | 29.7 | 31.3 | 22.0 | 28.8 | 34.1 | 43.9 | 46.3 | 51.5 |
| ITQ | 7.3 | 13.8 | 14.4 | 20.9 | 25.3 | 30.3 | 24.5 | 26.8 | 34.8 | 43.2 | 49.1 | 51.5 |
| SH | 12.1 | 14.2 | 17.5 | 20.0 | 26.4 | 36.0 | 19.7 | 24.8 | 32.9 | 38.7 | 45.4 | 46.7 |
| PCAH | 11.5 | 13.8 | 16.4 | 22.6 | 31.1 | 38.6 | 19.7 | 24.8 | 32.9 | 38.7 | 45.4 | 46.7 |
| | **RG** (78.7) | | | | | | **Mturk** (71.1) | | | | | |
| Ours | 24.0 | 40.4 | 51.3 | 62.3 | 63.2 | 55.8 | 44.0 | 49.0 | 52.2 | 60.1 | 57.7 | 55.2 |
| LSH | 25.5 | 24.9 | 34.6 | 62.1 | 61.8 | 73.5 | 18.3 | 31.3 | 31.4 | 42.9 | 56.5 | 60.7 |
| RandExp | 28.7 | 45.6 | 47.3 | 63.7 | 67.8 | 70.8 | 28.3 | 41.3 | 50.1 | 56.5 | 65.4 | 67.1 |
| ITQ | 21.4 | 32.7 | 50.4 | 57.7 | 67.6 | 70.3 | 26.3 | 41.4 | 53.2 | 61.2 | 67.1 | 68.9 |
| SH | 39.8 | 45.6 | 50.0 | 50.2 | 62.3 | 68.6 | 20.3 | 35.9 | 51.9 | 61.9 | 59.1 | 61.3 |
| PCAH | 45.0 | 50.0 | 49.2 | 46.8 | 66.6 | 69.8 | 24.9 | 40.7 | 55.7 | 64.3 | 64.4 | 60.5 |

Table 9: Evaluation on word similarity datasets, analogous to Table 7. The 300d GloVe embeddings trained from scratch on the same OpenWebText dataset as our algorithm.

is stored in the CPU memory. The parallelization strategy in our implementation is based on two aspects. First, each minibatch of data is divided into smaller sub-minibatches which are processed on different GPUs. Second, all the operations (dense-sparse matrix multiplications, arg max operation, and weight updates) are executed in parallel using multiple threads.

## 10 APPENDIX E. QUALITATIVE EVALUATION OF CONTEXTUAL EMBEDDINGS.

In order to evaluate the quality of contextualized embeddings we have created an online tool, which we are planning to release with the paper, that allows users to explore the representations learned by our model for various inputs (context-target pairs). For a given query the tool returns the word cloud visualizations for each of the four top activated Kenyon cells. We show some examples of the outputs produced by this tool in Fig. 6. Each query is used to generate a bag of words input vector $\mathbf{v^A}$. This vector is then used to compute the activations of KCs using $\left\langle \mathbf{W}_\mu, \mathbf{v^A} \right\rangle$. Top four KCs with the highest activations are selected. The corresponding four weight vectors are used to generate four probability distributions of individual words learned by those KCs by passing the weights through a softmax function. For example, for one of those vectors with index $\mu$, the probability distribution is computed as $\mathrm{prob}_i = SM(W_{\mu i})$. These probability distributions for the top four activated KCs are visualized as word clouds. In computing the softmax only the target block of the weight vector was used (we have checked that using only the context block gives qualitatively similar word clouds).

**Query:** Entertainment industry shares rise following the premiere of the mass destruction weapon documentary

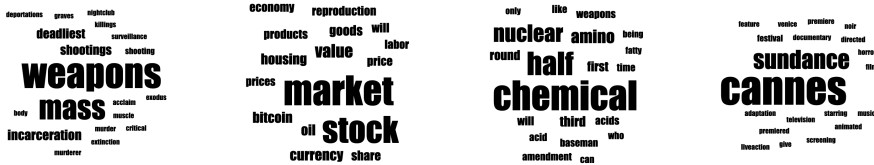

**Query:** European Court of Human Rights most compelling cases

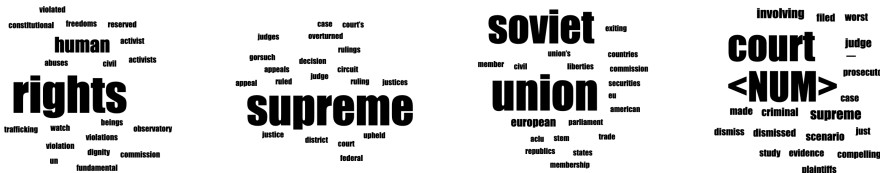

**Query:** Senate majority leader discussed the issue with the members of the committee

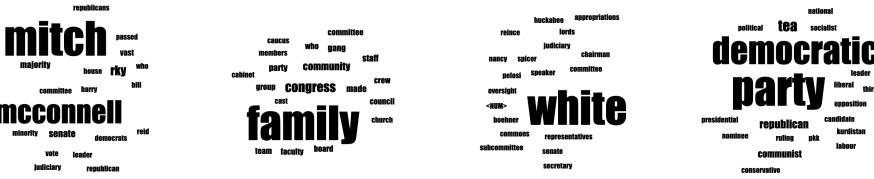

Figure 6: Examples of three queries and corresponding word cloud visualization for top four activated KCs (by each query).

The results indicate that the fruit fly network indeed has learned meaningful representations. Consider for example the first query. The sentence: "Entertainment industry shares rise following the premiere of the mass destruction weapon documentary" results in the four top activated KCs shown in Fig. 6. The top activated KC has the largest weights for the words "weapon", "mass", etc. The second activated KC is sensitive to the words "market", "stock", etc. This illustrates how the fruit fly network processes the queries. In this example the query refers to several distinct combinations of concepts: "weapon of mass destruction", "stock market", "movie industry". Each of those concepts has a dedicated KC responsible for it. As one can see the responses are not perfect. For example in this case one would expect to have the 4-th highest activated KC, which is responsible for the "movie industry" concept to have a higher activation than the 3-rd highest KC, which is responsible for the types of "weapons of mass destruction". But overall all the concepts picked by the KCs are meaningful and related to the query.

## 11 APPENDIX F. DETAILS OF GLOVE RETRAINING

To directly compare our method to GloVe, we trained an uninitialized GloVe model on the same OpenWebText corpus using the code provided by the original GloVe authors [34][4]. This model was optimized to have the same vocab size as our model (the 20k most frequent tokens), used an embedding size of 300, and a window size of 15. The model was trained for 180 iterations at about 3 minutes, 20 seconds per iteration on 16 threads, resulting in the total training time of approximately 10 hours.

## 12 ACKNOWLEDGEMENTS

We are thankful to L.Amini, S.Chang, D.Cox, J.Hopfield, Y.Kim, and H.Strobelt for help-ful discussions. This work was supported by the Rensselaer-IBM AI Research Collaboration (http://airc.rpi.edu), part of the IBM AI Horizons Network (http://ibm.biz/AIHorizons).

---

[4]https://nlp.stanford.edu/projects/glove/

