# OpenReview forum: "Can a Fruit Fly Learn Word Embeddings?"
_ICLR.cc/2021/Conference — ICLR 2021 Poster_

### Official Review · AnonReviewer4 · 2020-10-26
**Biologically inspired word embeddings with interesting results**

**Rating:** 7
**Confidence:** 4

**Review:**

Summary:
The authors present a formalization of a simple biological network (the mushroom body of the fruit fly), that allows very efficient “biologically inspired” word embeddings. They train this network to generate both static (context-independent) and context dependent embeddings, and evaluate these embeddings using several metrics comparing mainly to GloVe embeddings, and to some extent to BERT embeddings. Although the results are sometimes inferior, they are overall comparable, and importantly achieved at significantly lower computational resources. The main contribution of this work is not this specific network formalization (which is nice), but rather demonstrating that formalizing biological networks can generate more efficient algorithms, that achieve results comparable to the complex algorithms used ubiquitously.

Strengths:
+ The approach of the paper, namely generating word embeddings using a formalization of a well-known biological network, is interesting and inspiring
+ The authors evaluate the embeddings using both intrinsic and extrinsic evaluation methods, which is important
+ The learned embeddings are surprisingly strong, given the small and computationally efficient network. Furthermore, the embeddings seem to separate concepts well. These results support the idea that correlations between words and contexts, can indeed be extracted from raw text (in an unsupervised manner) by this network. I find this important because although the biological network of the fruit fly clearly did not evolve to perform NLP tasks, the fact that it evolved to process input from several modalities seems to allow it to learn efficient embeddings.

Weaknesses:
- I would have liked to see comparison of the resulting embeddings to more than GloVe. E.g. see https://arxiv.org/pdf/1901.09785.pdf, who discuss intrinsic and extrinsic evaluations of embeddings and compare to many embeddings. For example, GloVe is actually not the best for word similarity.
- For the extrinsic tasks describes in the document classification section, I would have liked to see a comparison to BERT as well, not just GloVe (or even other embeddings). I think these tasks can indeed use context.
- I think it would also be useful to evaluate on more extrinsic tasks, although given the space constraints I understand this may be difficult.
- Finally, it would have been good to see more analysis of the embeddings, e.g. by doing error analysis in one or more of the evaluation methods.

Recommendation:
I vote for accepting this paper. I very much like the approach of looking at biological networks and trying to learn from them how one can efficiently perform computations. I would have liked to see a more in-depth comparison and analysis in some places, but the results as-is are promising, interesting and inspiring enough to be of interest to the ICLR community.

Questions and minor comments:
1. In Figure 2, the legend reads to me as if the context words are in light blue, but in the figure both target word and context words are in light blue.
2. I think it would be helpful to make a comment that context in this paper is essentially Bag Of Words, as there is no positional information.
3. In the semantic similarity section, the authors refer to the supplement for binarized GloVe embedding results. I looked at the supplemental results and I found them compelling. I recommend at least saying the conclusion in the main paper (that your binary word embeddings are competitive with binarized GloVe embeddings), rather than just sending the reader to the supplement.

---

> ### Author Response · Authors · 2020-11-24
> **Response to AnonReviewer4**
>
> Thanks for your enthusiastic feedback and nice comments! We address the questions/comments below:
>
> 1. (I would have liked to see comparison of the resulting embeddings to more than GloVe): We have also added comparison with word2vec embeddings, please see column 4 of Table 1 and Table 8 in the Supplemental. Please see also added comparisons with word2vec in Tables 3,4,5. And also an added comparison with Word2Sense in Tables 3,4 (per request of Rev 3).
> 2. (For the extrinsic tasks describes in the document classification section, I would have liked to see a comparison to BERT as well, not just GloVe, or even other embeddings): We have updated Table 5 with the results for word2vec, BERT, and NLB embeddings (Tissier et al., 2019).
> 3. (I think it would also be useful to evaluate on more extrinsic tasks): We have added two additional datasets for document classification to Table 5. They are WOS-11967(Kowsari et al., 2017) and TREC-6datasets (Li & Roth, 2002). Although they are the same tasks - document classification, these additional results extend our analysis on extrinsic evaluations by adding more datasets. Furthemore, for a fair comparison, we evaluate all methods on the same vocabulary (20k words), which removes variability of the results due to the (arbitrary) choice of the vocabulary. This explains the slightly updated numbers in Table 5.
> 4. (It would have been good to see more analysis of the embeddings, e.g. by doing error analysis in one or more of the evaluation methods): We have created an online tool, which we are planning to release with the paper, that allows users to explore the representations learned by our model for various inputs (context-target pairs). For a given query the tool returns the word cloud visualization for each of the four top activated Kenyon cells.  We have added Appendix D (section 9 ) to the Supplemental where we included some of those examples, please see Fig. 6.  As you can see, the model indeed nicely captures contextual meaning and polysemy.
> 5. (In Figure 2, the legend reads to me as...): Fixed!
> 6. (I think it would be helpful to make a comment that context in this paper...): Comment added in text referencing Fig. 2.
> 7. (In the semantic similarity section, the authors refer to the supplement for binarized GloVe...): Conclusion added for pretrained GloVe and also for word2vec and GloVe trained on OWT (Tables 7,8,9). We have also added Table 2 and explanation to the main text in sec 3.1 to briefly summarize those results.

---

### Official Review · AnonReviewer1 · 2020-10-27
**The method is interesting but the experimental evidence is not good enough**

**Rating:** 7
**Confidence:** 3

**Review:**

Although the paper does not say so, my understanding is that the proposed word embedding method actually first perform Kmeans-like clustering on the context vector shown in Figure 2. In the binary word embedding of each word, we set the dimensions corresponding to the k closest cluster centers to 1 and 0 otherwise. Most parts of the paper are describing how the simple word embedding method is related to the neural system of a fruit fly and showing the method achieves comparable performance compared with GloVe in word similarity tasks, context-sensitive word similarity datasets, and document classification tasks.

Pros:
1. It is interesting to see the connection between a biological neural system and word embedding.
2. As far as I know, the method is novel.
3. Such a simple and biologically plausible method could reach reasonable performance.

Cons:
1. The experiments have not demonstrated that the method is a useful word embedding method for NLP researchers. Some of the experiment designs, comparisons, and presentations are misleading.
2. Some related work and comparisons are missing.
3. The authors actually do not explain why this method works well in machine learning perspectives (similar to a biological neural system is not a very strong explanation for many people).

Clarity:
The paper is easy to understand, but it does not explain why the method works from the ML perspective.

Originality:
As far as I know, the method is novel.

Significance of this work:
This might have a large impact on the neuroscience field.

The method is compared with GloVe most of the time. First, GloVe actually does not perform well in word similarity tasks compared with word2vec/SGNS [3]. In terms of document classification, the results are mixed and I think the authors should try more datasets to know which method is better. In addition, GloVe is trained on a different dataset, so the result is not very comparable. Second, GloVe is a dense method. The proposed method should be compared with binary word embedding or compressed word embedding methods such as [1,2]. It will be even better if you can show that it is better than sparse representation such as PPMI in [3], even though PPMI is not binary and uses much more dimensions (you might want to use the dot product values instead of binary value in Equation (4) to improve your method and make the comparison fairer).

As I mention at the beginning, I believe the word embedding is actually a binary projection to Kmeans-like cluster centers in the context vector space. Here is why. Let's first assume W_{mu} and v^A are normalized and ignore the normalization term /p to build the connection. Then, the l2 distance is (1 - dot product). Equation (1) tells us that we are minimizing the l2 distance between v^A and its closest cluster center (i.e., W_{mu}). Finally, Equation (4) clearly describes the top k projection. If I am right, I think adding this perspective will make readers understand the method better. If I am wrong, please explain why and try to explain what the method is actually doing in some machine learning perspectives.

If I am right, we will find the description section 3.2 might be misleading. The main reason that the average cosine similarity within the cluster is higher might be because the binarization projection makes the word embeddings locally collapse to small cluster centers, which makes the word embeddings cannot capture the fine-grained statistics details. You can visualize the word embedding in a 2D space to see if I am right. If I am right, this is not an advantage in my opinion.

If the authors could show that the proposed method is state of the art compared with binary word embeddings in a fair setting (e.g., train on the same dataset), I will vote acceptance. Otherwise, I am very likely to keep my weak rejection vote.

Minor:
1. The definition of p before Equation (1) is unclear. Why do you have a mod here?
2. The optimization method in Equation (2) is unclear. What does dt mean? Is t the time step in iterative optimization? Is this a kind of gradient descent? I guess the differential equation comes from neural science and it needs more explanations to make NLP researchers understand. It would be better if the author can compare with other optimization methods such as gradient descent and EM algorithm for Kmeans clustering.


[1] Tissier, Julien, Christophe Gravier, and Amaury Habrard. "Near-lossless binarization of word embeddings." Proceedings of the AAAI Conference on Artificial Intelligence. Vol. 33. 2019.
[2] Wang, Yuwei, et al. "Biological Neuron Coding Inspired Binary Word Embeddings." Cognitive Computation 11.5 (2019): 676-684.
[3] Levy, Omer, Yoav Goldberg, and Ido Dagan. "Improving distributional similarity with lessons learned from word embeddings." Transactions of the Association for Computational Linguistics 3 (2015): 211-225.

---

> ### Author Response · Authors · 2020-11-24
> **Response to AnonReviewer1**
>
> Thank you for your feedback, we are glad to hear that you find the connection between the biological neural system and word embedding interesting. Below we address the questions raised:
>
> 1. (Why this method works well in machine learning perspective): We have added a paragraph at the end of section 2.2 to explain the relationship of our method to spherical K-means and provide some ML intuition about the algorithm.
> 2. (Comparison with more than GloVe): We have also added comparison with word2vec embeddings, please see column 4 of Table 1 and Table 8 in the Supplemental.
> 3. (In terms of document classification, the authors should try more datasets…) We have added two additional datasets: WOS-11967(Kowsari et al., 2017) and TREC-6datasets (Li & Roth, 2002) to Table 5.
> 4. (GloVe is trained on a different dataset): We have trained GloVe on OpenWebText (the corpus that we used for training our algorithm). The results are reported in Table 9 in the Supplemental, technical details of training are reported in Appendix E (section 10). We have constrained the vocabulary to be the same as the vocabulary of our method. The overall wall-clock training time was approximately 10 hours, which is significantly larger than the training time for our method (see Table 6).
> 5. (Method should be compared with binary word embedding or compressed word embedding) We did compare with binary word embedding in the initial submission. Table 7 in Appendix B is dedicated to this comparison. In that table, NLB refers to the method of Ref [1] that you recommended. We have also created a similar table, Table 8, to compare with common binarization methods of word2vec, and Table 9 with comparisons to GloVe trained on OWT. We have also added Table 2 to the main text with a short summary of these results for hash length k=4.
> 6. (Word embedding is a binary projection to Kmeans-like cluster centers): This is only partly correct. We have added a  discussion of the relationship  between K-means and our algorithm at the end of section 2.2. Indeed, for constant $p_i$ our algorithm is a version of spherical K-means clustering. More precisely it is a biologically plausible implementation of spherical K-means clustering. But this is not the case for generic $p_i$, as we explain in the last paragraph of section 2.2. We hope this answers your question.
> 7. (Description section 3.2 might be misleading…): We have conducted additional experiments by binarizing pretrained GloVe embeddings at varying thresholds as well as comparing with other binarization methods (Tissier at al.) and plotted all of them on the diagram in Fig.3. The conclusion is that the improvement in clustering quality (compared to continuous GloVe) is indeed due to binarization. As seen in Fig. 3, panel A, the clusters lose detail (i.e., both intra- and inter-cluster similarity increases) as the binarization threshold gets higher (shown for Glove). Our embeddings maintain a balance between intra- and inter-clustering similarity, and thus still capture fine-grained cluster information.  Thanks for the suggestion to do these additional experiments.
> 8. (If the authors could show that the proposed method is SOTA compared with binary word embeddings in a fair setting, e.g., train on the same dataset, I will vote acceptance): We include results for GloVe trained using the same dataset (OWT) and the same vocabulary as for our method. Word similarity analysis results are reported in Table 9 of the Supplemental. Please notice that fruit fly embeddings are SOTA for small hash lengths. Specifically, our method outperforms existing alternatives for binarization on MEN and WS353 (for $k=4,8,16$), SIMPLEX (for $k=4$), RG (for $k=16$), and Mturk (for $k=4,8$). Small hash lengths are particularly valuable for compression, since the memory footprint grows with the hash length (it is sufficient to store only the indices of the entries that are assigned state 1).
> 9. (The definition of p before Eq (1) is unclear. Why do you have a mod here?): Please see footnote 1 that we added to the revised paper to explain this.
> 10. (The optimization method in Eq (2) is unclear. What does dt mean?...): We have changed the notations in Eq 2 to be more consistent with machine learning literature. Regarding alternative ways of doing optimization, the right hand side of Eq (2) does not coincide with the gradient of the energy function (1). However, as explained in (Ryali et al., 2020), updating the weights using Eq (2) always leads to decreasing the energy. We have done an extensive set of experiments on simple datasets (MNIST) to check that the two optimization procedures - naive gradient descent on energy (1) vs. equation (2) - obtain equally good solutions with approximately the same final value of energy. The advantage of using Eq (2) compared to the naive gradient descent is that it satisfies the constraints of biological plausibility, which is important for our approach since we are modelling a biological network.

---

> > ### Comment · AnonReviewer1 · 2020-11-24
> > **The revision is strong**
> >
> > I think almost all of my concerns are resolved.
> > I will change my vote to acceptance.
> > (by the way, sorry that I miss your appendix when I review the paper)

---

> > > ### Author Response · Authors · 2020-11-24
> > > **Thanks**
> > >
> > > Thank you! We are glad you like our paper, and we appreciate your insightful comments!

---

### Official Review · AnonReviewer3 · 2020-10-29
**a biological network inspired model that learns unsupervised word embeddings**

**Rating:** 7
**Confidence:** 4

**Review:**

This paper presents a simplified model of a biological system, and asks the question of whether it can learn unsupervised word embeddings. The objective function and its optimization are clear and unreasonable. (Note: I do not have the expertise to comment on whether this is an accurate characterization of the working of the biological system.) The authors connect the optimization problem to other recent learnings in this area such as sparse learnt projection -- this perspective is appreciated.

To establish the quality of the embeddings, the authors compare them to GloVe and BERT embeddings on word specific tasks like similarity and contextual tasks on datasets like SCWS. In these comparisons, the quality of embeddings is relatable to GloVe, and inferior to BERT. There is also a small qualitative demonstration of the embeddings (Fig. 3) This tells us that the embeddings are meaningful, and their quality is comparable to Glove, if not the newer generation embeddings. I think this is a good outcome for the model they develop.

 For these reasons, I am generally in favor of accepting the paper. However, I have reservations about particular aspects, and request the authors to edit the following content.
1. SOTA claims. GloVe is a great candidate to compare a new methodology to, but is not SOTA for unsupervised embeddings. There are many other unsupervised non-DL methods (e.g. Word2vec) which do better than GloVe on specific tasks. It is important to clarify and cite appropriate SoTA for each task as of the time of writing.

2. I am not sure the comparison to GloVe for contextual evaluation is the most appropriate. There are other works (e.g. Word2Sense, ACL'19 and citations therein) for stronger contextual embeddings. Please bechmark against this.

3. I feel the comparison with BERT is mostly irrelevant. I think comparison with GloVe and related work are sufficient to show that the new methodology has something going for it. I would not have expected it to compete with BERT like models nor serve all their use cases. Therefore, claims that training is faster than BERT are also mostly unnecessary.

4. I would instead place more emphasis on qualitive and quantitative evidence that the embeddings capture contextual meanings, polysemy etc. Consider expanding section 3.2 with more evidence.

With these changes, I would make a stronger recommendation.

Revised rating to 7 after revision.

---

> ### Author Response · Authors · 2020-11-24
> **Response to AnonReviewer3**
>
> Thank you for your feedback! We agree that it’s a good outcome that the embeddings generated by this biological model are comparable with GloVe. Below we address the concerns:
>
> 1. (SOTA claims and comparison with word2vec): We have added references to SOTA results throughout the manuscript. Particularly, please see the last column of Table 1. We have also added comparison with word2vec embeddings, please see column 4 of Table 1 and Table 8 in the Supplemental.  We have changed how we present the results of our algorithm in Table 1 of the main text, which summarizes  the results of Table 7 (identical to the initial submission) and Table 8. In Table 1 we choose a single hash length for all the evaluations instead of reporting the best result on each dataset, resulting in slightly updated numbers in Table 1.
> 2. (Word2Sense):  For context-dependent embeddings we have compared with Word2Sense in Tables 3 and 4, and we also included word2vec results. Word2Sense performs similarly to Glove on SCWS, but is indeed better on WiC (which is more context dependent). However, our approach is still better than Word2Sense and word2vec on WiC. Thanks for the recommendation to do these experiments!
> 3. (I feel the comparison with BERT is mostly irrelevant. I think comparison with GloVe and related work... ): We partly agree with you, however we chose to keep those results in the manuscript, particularly given that other reviewers emphasized their importance and requested to expand them.
> 4. (I would instead place more emphasis on qualitative and quantitative evidence that the embeddings capture contextual meanings, polysemy etc.): We have created an online tool, which we are planning to release with the paper, that allows users to explore the representations learned by our model for various inputs (context-target pairs). For a given query the tool returns the word cloud visualization for each of the four top activated Kenyon cells.  We have added Appendix D (section 9 ) to the Supplemental where we included some of those examples, please see Fig. 6.  As you can see, the model indeed nicely captures contextual meaning and polysemy.

---

> > ### Comment · AnonReviewer3 · 2020-11-24
> > **Comments on revision**
> >
> > Thank you for the revisions. Re. point 2: WordCtx2Sense is the best candidate for contextual embeddings in the Word2Sense paper. Please consider comparing against that instead of Word2Sense in your Tables 3 and 4. Apart from that, the update looks good to me.

---

> > > ### Author Response · Authors · 2020-11-25
> > > **Thank you! We will do our best to compare with WordCtx2Sense.**
> > >
> > > Thank you for your time to evaluate our response and for the suggestion to compare with WordCtx2Sense! We compared to Word2Sense since the pretrained embeddings were available on their github directory (https://github.com/abhishekpanigrahi1996/Word2Sense). However, the WordCtx2Sense embeddings are not available there, and therefore we will have to retrain their model to obtain them.  We will do our best to add this comparison to the final version of our paper.

---

### Author Response · Authors · 2020-11-24
**General Response**

We thank all the reviewers for the valuable feedback and many positive comments! We have expanded the evaluations with comparisons to word2vec, GloVe trained on OpenWebText, various common binarization techniques, Word2Sense, and added additional datasets. These updated results  appear in the corresponding tables in the revised paper. For example, Table 5 evaluates new datasets and more methods (for extrinsic evaluations), and expands/merges Tables 4 and 5 of the initial submission. We have moved a paragraph “Related Work” to Appendix F in the Supplemental (because of space constraints). We have also added an Appendix D (section 9 of the Supplemental) where we expanded on the qualitative evidence that the fruit fly network learns meaningful representations, which we have visualized using word clouds (please see Fig. 6).

The conclusions remain unchanged. The fruit fly network learns binary word embeddings that capture semantic meanings of individual words, as well as context-dependency. The resulting binary embeddings produce competitive results on both intrinsic and extrinsic tasks, and require a smaller amount of computational resources than the classical methods.

---

### Decision · Program_Chairs · 2021-01-07
**Final Decision**

**Decision:**

Accept (Poster)

**Comment:**

This work is likely to lead to more connections between machine learning and neuroscience at a fine-grained level where ML methods can help explain and understand neural circuits.

To encourage this, it would be helpful if authors described the biology of the PN-KC-APL network and the known constraints over possible formalizations of that network. The authors present one formalization, but little discussion is given toward the design space for such models. Are there other possible ways to describe the PN-KC-APL network? Are all alternate ways to do so equivalent to the model presented here? What properties are unknown and how could they affect the formalization presented here?

Overall, reviewers agree this is a good submission.